# OsACA9, an Autoinhibited Ca^2+^-ATPase, Synergically Regulates Disease Resistance and Leaf Senescence in Rice

**DOI:** 10.3390/ijms25031874

**Published:** 2024-02-03

**Authors:** Xinyu Wang, Ziyao Wang, Yiduo Lu, Jiani Huang, Zhuoer Hu, Junlei Lou, Xinyue Fan, Zhimin Gu, Pengcheng Liu, Bojun Ma, Xifeng Chen

**Affiliations:** College of Life Sciences, Zhejiang Normal University, Jinhua 321004, China; Xinyu_KT@126.com (X.W.); lyd_18806777577_48@126.com (Y.L.); huze0222@163.com (Z.H.); jllou1026@163.com (J.L.); zhmgu@zjnu.cn (Z.G.); liupc2019@zjnu.edu.cn (P.L.);

**Keywords:** rice (*Oryza sativa* L.), Ca^2+^-ATPase, *OsACA9*, disease resistance, leaf senescence

## Abstract

Calcium (Ca^2+^) is a versatile intracellular second messenger that regulates several signaling pathways involved in growth, development, stress tolerance, and immune response in plants. Autoinhibited Ca^2+^-ATPases (ACAs) play an important role in the regulation of cellular Ca^2+^ homeostasis. Here, we systematically analyzed the putative OsACA family members in rice, and according to the phylogenetic tree of OsACAs, OsACA9 was clustered into a separated branch in which its homologous gene in *Arabidopsis thaliana* was reported to be involved in defense response. When the *OsACA9* gene was knocked out by CRISPR/Cas9, significant accumulation of reactive oxygen species (ROS) was detected in the mutant lines. Meanwhile, the *OsACA9* knock out lines showed enhanced disease resistance to both rice bacterial blight (BB) and bacterial leaf streak (BLS). In addition, compared to the wild-type (WT), the mutant lines displayed an early leaf senescence phenotype, and the agronomy traits of their plant height, panicle length, and grain yield were significantly decreased. Transcriptome analysis by RNA-Seq showed that the differentially expressed genes (DEGs) between WT and the *Osaca9* mutant were mainly enriched in basal immune pathways and antibacterial metabolite synthesis pathways. Among them, multiple genes related to rice disease resistance, *receptor-like cytoplasmic kinases* (*RLCKs*) and *cell wall-associated kinases* (*WAKs*) genes were upregulated. Our results suggest that the Ca^2+^-ATPase OsACA9 may trigger oxidative burst in response to various pathogens and synergically regulate disease resistance and leaf senescence in rice.

## 1. Introduction

In plants, Ca^2+^ signaling is involved in all aspects of life, including growth regulation, development, reproduction, stress responses, and the establishment of beneficial plant-microbe interactions [1]. Ca^2+^ influx is the signal of the earliest responses stimulated by pathogen/microbe-associated molecular patterns (PAMPs/MAMPs) and functions in stomatal immunity [2,3]. Upon elicitor perception, the initial activation of RESPIRATORY BURST OXIDASE HOMOLOGUE (RBOH) primes the system for subsequent activation through Ca^2+^ signaling, leading to reactive oxygen species (ROS) production, which further enhances the immunity [2,4]. It is also a vital signal in response to abiotic stresses, such as touch, cold, drought, hypoxia, and salinity [5,6,7]. In addition, Ca^2+^ participates in the regulation of polarized cell growth, the elongation of pollen tubes and root hairs, and plant senescence [1,8]. Proper Ca^2+^ concentration could inhibit chlorophyll and protein degradation to inhibit leaf senescence, while an excessive Ca^2+^ influx can induce the production of reactive oxygen species (ROS) and promote aging [9].

Ca^2+^-ATPases belong to the superfamily of P-type ATPases forming phosphor-aspartate enzyme intermediates during the reaction cycle [10]. They play an important role in maintaining Ca^2+^ homeostasis in plant cells by controlling Ca^2+^ efflux from the cytosol to organelles and/or to the apoplast [11]. Ca^2+^-ATPases have four highly conserved protein domains, including a cation transporter/ATPase N-terminal (PF00690), a cation transporter/ATPase C-terminal (PF00689), an E1–E2 ATPase (PF00122), and a haloacid dehalogenase-like hydrolase (PF00702) [12]. According to their protein sequences and protein locations in plants, plant Ca^2+^-ATPases can be grouped into two phylogenetic subgroups, P-type IIA ER-type Ca^2+^-ATPases (ECAs) and P-type IIB autoinhibited Ca^2+^-ATPases (ACAs) [13]. ACAs are characterized by an N-terminally situated calmodulin (CaM)-binding domain (CMBD). Autoinhibition can be relieved by Ca^2+^/CaM-binding, thus allowing ACAs to be directly controlled by free Ca^2+^ levels and giving rapid feedback regulation of Ca^2+^ signals [14,15]. 

To date, *ACA* gene family members have been identified in multiple species. Ten *AtACA* genes were first identified in *Arabidopsis thaliana* [16]. Subsequently, *ACA* gene families have been reported in *Solanum lycopersicum* [17], *Nicotiana tabacum* [17], *Capsicum annuum* [17], *Solanum tuberosum* [17], *Brassica rapa* [18], *Ricinus communis* [19], *Zea mays* [20], and *Triticum aestivum* [21]. *ACA* genes have been proven to function in the development and growth of plants [11]. For instance, *AtACA1* regulates stomatal aperture, cytosolic distribution of chloroplasts in response to light, and root gravitropic curvature [22]. *AtACA2*, *AtACA7*, *AtACA9*, and *AtACA13* contribute to pollen germination and fitness [23,24,25]. *AtACA10* is involved in vegetative growth and inflorescence structure [26]. Further, *ACA* genes also play important roles in biotic stress responses in plants [27]. *AtACA2* and *AtACA4* can alleviate hypersensitivity to salt [28]. *GmSCA1* is highly and rapidly induced by salt [29]. *GsACA1* positively regulates plant tolerance to both salt and carbonate-alkaline stresses in *Glycine soja* [30]. In moss (*Physcomitrella patens*), the knockout of the *ACA* gene *PCA1* results in an increased sensitivity to salt stress [31]. *TaACAs* are suggested to have diverse functions in various biological processes and stress responses [21]. Recently, more specific evidence suggested that *ACA* genes function in defense responses in plants. *GmSCA1* is induced by a fungal elicitor [29]. *AtACA11* is a genetic suppressor of the programmed cell death pathway in Arabidopsis [32]. *AtACA4*, *AtACA8*, and *AtACA11* mediate the pathogen effector flg22-dependent Ca^2+^ signaling [33,34]. AtACA8 interacts with FLAGELLIN SENSITIVE2 (FLS2), the receptor for flg22, along with AtACA10 and AtACA12 [35]. 

In rice, *OsACA* gene family members have been identified by bioinformatics methods [36,37]. *OsACA1* and *OsACA8* are reported to be early-glutamate-responsive in rice roots [38]. *OsACA1* is positively regulated by OsTF1L, a rice HD-Zip transcription factor which promotes lignin biosynthesis and stomatal closure that improves drought tolerance [39]. *OsACA6* can efficiently modulate the ROS metabolism and proline biosynthesis in response to the abiotic stresses of salt, drought, and heat in tobacco [40,41]. *OsACA7* is downregulated upon the supplementation of Silicon (Si) under abiotic and biotic stresses [42,43]. Currently, the knowledge of *OsACAs* is fragmented, and is mainly focused on stress responses and the regulation of plant growth and development; solid evidence on the role of *OsACAs* in biotic responses is still limited.

Here, we focused on an especially distinctive clade of *OsACA* genes, based on a phylogenetic tree between Arabidopsis (*A. thaliana*), Tomato (*S. lycopersicum*), maize (*Z. mays*), and rice (*Oryza sativa*), which was reported to be related to defense response. We constructed two knockout lines of the *OsACA9* gene by CRISPR/Cas9 and characterized the phenotype and resistance spectrum of the knockout mutant. We carried out transcriptome analysis by RNA-Seq to investigate the molecular mechanism of defense response mediated by Ca^2+^-ATPase OsACA9 in rice. Our results suggest a pleiotropic effect of *OsACA9* on disease resistance and leaf senescence and provide new evidence for the role of Ca^2+^-ATPases in plant–pathogen interactions.

## 2. Results

### 2.1. Evolution Analysis and Expression Profile Suggest the Potential Role of OsACA9 in Various Biological Processes

Previously, eleven *OsACA* genes had been initially reported from rice [36,37]. However, one of the family members (*OsACA10*) was discarded due to the lack of the cation transporter/ATPase N-terminal domain and the E1–E2 ATPase domain [37]. To analyze the evolution of *OsACA* genes, a phylogenetic tree of ACA family proteins from rice (10 OsACAs), Arabidopsis (10 AtACAs), tomato (8 SlACAs), and maize (16 ZmACAs) was constructed and classified into four clades (Figure 1A). Interestingly, OsACA9 and OsACA11 were grouped with AtACA8 and AtACA10 in the same branch (Clade IV), which is especially distinctive for its role in defense response [35,44]. Moreover, the gene structure of *OsACA9* and *OsACA11* were similar, with multiple introns and exons, which was obviously different from other *OsACA* family members (Figure 1B). These results indicated that *OsACA9* and *OsACA11* were likely to be paralogous in evolution and might be redundant in biological function. Therefore, the expression level of *OsACA9* and *OsACA11* in different tissues of rice plants were analyzed by reverse transcription-quantitative real-time PCR (RT-qPCR). The results showed that *OsACA9* is highly expressed in various tissues in rice, especially in the leaf and stem, while OsACA11 was not or lowly expressed in all tissues tested (Figure 1C). These results suggest that OsACA9, as the homologue to AtACA8 and AtACA10, might biologically play a more important role in defense response as well as the growth and development in rice.

To investigate the function of *OsACA9* in rice, two sgRNAs (sgRNA-1, sgRNA-2) were designed (Figure 2A) and constructed into a CRISPR-Cas9 vector, respectively. The result vectors were genetically transformed into a wild-type rice cultivar Zhonghua 11 (ZH11). Two different transgenic lines, *aca9#1* and *aca9#2*, were identified, in which a thymine (T) base was inserted by the sgRNA-1, and four bases/one base were deleted by the sgRNA-2, respectively (Figure 2B). The insertion and deletion both resulted in reading frameshift mutations (knockout) and the early termination of translation of the *OsACA9* gene, and the two mutants of *OsACA9* were subsequently used for phenotype identification.

### 2.2. Knockout of OsACA9 Broadly Enhanced the Bacterial Disease Resistance of Rice

To identify the disease resistance of the *Osaca9* mutants, different strains of bacterial leaf blight (BB) and bacterial leaf streak (BLS) were used for analysis. Six *Xanthomonas oryzae* pv. *oryzae* (*Xoo*) strains of BB (PXO86, PXO71, POX99A, Zhe173, C2 and C5) and two *Xanthomonas oryzae* pv. *oryzicola* (*Xoc*) strains of BLS (BLS256 and RS105) were inoculated in the leaves of the *Osaca9* mutants and the wild-type control ZH11 (susceptible cultivar), respectively. The results showed that the lesion lengths of all the *Xoo* and *Xoc* strains formed in the leaves of the *Osaca9* mutants were significantly decreased compared to the wild-type control (Figure 3A–C). This result suggested that the knockout of *OsACA9* could broadly enhance the resistance of rice plants to bacterial disease. Moreover, it was found that the major ROS, hydrogen peroxide (H_2_O_2_), was excessively accumulated in the leaves of the *Osaca9* mutants. The histological staining using 3,3′-diaminobenzidine (DAB) showed abundant brown spots in the leaves of the *Osaca9* mutants, especially at leaf margins (Figure 3D), and the content of H_2_O_2_ was higher than that of the wild-type control (Figure 3E). The result suggested that the mutation of *OsACA9* caused accumulation of ROS in leaves, which may contribute to its enhanced resistance to pathogens.

### 2.3. OsACA9 Regulated the Leaf Senescence and Development of Rice

Based on current knowledge, ROS has been proved a driving force for the accelerated senescence of leaves [45,46]. At the filling stage, the leaves of *Osaca9* mutants appeared yellowish, and the plant height significantly decreased (Figure 4A,B and Figure 5F). The net photosynthetic rate (*Pn*) of flag leaves in *Osaca9* mutants was significantly lower than that of the wild-type control (Figure 4C). The chlorophyll a (Chla), chlorophyll b (Chlb), and carotenoid (Car) contents of the flag leaves, second top leaves, and third top leaves from a single tiller of *Osaca9* mutants were significantly reduced (Figure 4D). All these results indicated that the *Osaca9* mutants exhibited an early-leaf-senescence phenotype and reduced photosynthesis efficiency in leaves.

Additionally, the agronomic traits of the *Osaca9* mutants and the wild-type control were assayed at the mature stage. For the *Osaca9* mutants, the flag leaf length (Figure 5D) and width (Figure 5E), the panicle length (Figure 5A,G), the grain number per panicle (Figure 5B,H), and the grain thickness (Figure 5M) were significantly decreased; however, the effective tiller number per plant (Figure 5I) and the grain length (Figure 5C,K) were significantly increased. There was no significant difference in seed setting rate (Figure 5J), the grain width (Figure 5L), and 1000-grain weight (Figure 5N) between the *Osaca9* mutants and the wild-type control. As a result, the yield per plant of the *Osaca9* mutants was decreased by about 15% of that of the wild-type control (Figure 5O).

### 2.4. Transcriptomic Analysis Suggested OsACA9 Involved in MAPK Cascades in PAMP-Triggered Defense Response

To investigate the potential molecular mechanism of the *OsACA9* gene involved in regulating disease resistance and early leaf senescence, total RNAs from wild-type control and the *Osaca9* knockout lines were respectively extracted and used for the deep sequencing by RNA-Seq technology. A total of 291 differentially expressed genes (DEGs; Log2 radio ≥1 or ≤−1; *p* < 0.05; Appendix A) were identified, among which 234 genes were upregulated and 57 genes were downregulated. GO (gene ontology) enrichment analysis showed that the DEGs were significantly enriched in molecular functions related to binding to organic compounds, ATP and ions, and kinase activity. Further, DEGs were also enriched in biological processes that respond to external abiotic stimuli (Appendix A). 

Moreover, the KEGG (Kyoto Encyclopedia of Genes and Genomes) pathway enrichment analysis of the DEGs revealed that these genes were mainly involved in disease resistance pathways including MAPK signaling and plant–pathogen interactions, both of whose function in plant immunity have been extensively studied and supported by solid evidence [47]. Meanwhile, multiple pathways related to basal metabolism and immune metabolites were also significantly enriched, such as the alpha-Linolenic acid metabolism pathway; the glycine, serine and threonine metabolism pathway; and the biosynthesis of terpenoids and fatty acid as well (Figure 6A). Among those DEGs, five *receptor-like cytoplasmic kinases* (*RLCKs*) genes and two cell *wall-associated kinases* (*WAKs*) genes were upregulated, which are central players in basal immunity [47]. Moreover, several functionally known genes related to rice disease resistance were also upregulated, including rice sheath blight (RSB), disease resistance gene *OsRSR* (*disease resistance protein RPM1*), *MDPK* (*malectin domain protein kinase*), and rice blast disease resistance associated gene *OsRBBI2* (*Bowman–Birk trypsin inhibitor*). Interestingly, *OsSULTR3;6* encoding a sulphate transporter in rice, which is considered as the key susceptibility gene to BLS [48], was downregulated (Appendix A). At the same time, the amounts of genes involved in the synthesis of disease resistant metabolites were also significantly upregulated, such as the 4-coumarate-CoA ligase OsACS6, the aldehyde oxidase OsAO2, the 1-aminocyclopropane-1-carboxylate oxidases Os2ODD8 and Os2ODD23, and the beta-sesquiphellandrene synthases OsTPS30 and OsTPS31. In addition, several transcription factors (TFs) involved in disease resistance were also upregulated (Figure 6B, Table 1, Appendix A). Gene expression assessed by RT-qPCR in the wild type and two lines of *Osaca9* was consistent with that measured by transcriptome sequencing (Figure 6C). Thus, OsACA9 might enhance disease resistance by activating the basal immune pathways and antibacterial metabolite synthesis pathways simultaneously.

## 3. Discussion

The *ACA* gene family is identified to be involved in multiple biological process, especially in stress responses. Based on the expression profile analysis, the expression level of *ACA* genes varies under abiotic stresses of salt, drought, and heat in multiple species [21,27,28,40,41]. However, recently, mounting evidence shows that *ACAs* have an additional role in defense response in *Arabidopsis* [34,44], but none of the studies have reported *ACA’s* role in disease resistance in rice yet. According to the results of phylogenetic analysis between *O. sativa*, *A. thaliana*, *S. lycopersicum,* and *Z. mays* (Figure 1A), we focused on an especially distinctive clade (Clade IV) related to biotic response including AtACA8 and AtACA10 [35,44,68], while other clades are mainly involved in the regulation of plant growth, development, and abiotic stress responses [11,17]. We constructed two knockout lines of the *OsACA9* gene by CRISPR/Cas9 and characterized the phenotype and resistance spectrum of the knockout mutants. We carried out transcriptome analysis by RNA-Seq to investigate the molecular mechanism of defense response mediated by Ca^2+^-ATPase OsACA9 in rice. Our results suggest a pleiotropic effect of *OsACA9* on disease resistance and leaf senescence and provide new evidence for the role of Ca^2+^-ATPases in plant–pathogen signaling.

Based on current knowledge, AtACA8 and AtACA10 were reported to be involved in PAMP-mediated immune response and interact with the receptor kinase FLS2 [44,68]. Naturally, we expected *Osaca9* to show the similar sensitive phenotype to pathogens as *aca8* and *aca10* mutants. However, the resistance evaluation results turned out to be absolutely the opposite, showing that two knockout lines of *Osaca9* were more resistant to multiple strains of *Xoo* including PXO86, PXO71, POX99A, Zhe173, C2, and C5, and *Xoc* strains including BLS256 and RS105 (Figure 3A–C). The transcriptome analysis showed that DEGs were enriched in pathways including MAPK signaling and plant–pathogen interactions (Figure 6A). Many of the DEGs encoded disease resistance proteins, growth/development-related proteins, signaling components, and transcription factors, as well as proteins involved in protein phosphorylation (Figure 6B; Table 1). Among them were multiple genes encoding RLCKs, which have emerged as a major class of signaling proteins that regulate plant cellular activities in response to biotic stresses and endogenous extracellular signaling molecules. By associating with immune receptor kinases (RKs), RLCKs regulate multiple downstream signaling nodes to orchestrate a complex array of defense responses against microbial pathogens [47]. Interestingly, the receptor for flg22, FLS2, was upregulated, while the brassinosteroid insensitive 1–associated kinase 1 (BAK1), an important factor in BRI1-mediated BR signaling [69,70], was downregulated. These results suggest that OsACA9 may have a similar signal transduction mechanism to AtACA8 and AtACA10 by affecting RKs and RLCKs in activating immune responses. Moreover, except for BB and BLS, multiple genes positively regulating rice resistance to sheath blight (SHB) and rice blast including *MDPK*, *OsWRKYs*, *OsbHLH057*, *OsRSR*, and *OsRBBI2* were upregulated (Table 1), which suggests the potential function of *OsACA9* in broad-spectrum resistance to pathogens. 

Meanwhile, the accumulation of ROS, an essential role in signaling plant immunity, was detected in the *Osaca9* mutants (Figure 3D,E). Interestingly, among all DEGs, *OsPIP2;7* encoding an aquaporin and *OsPRX7* encoding a peroxidase precursor, which function in ROS scavenging [35,40,71], were both upregulated (Figure 6B, Table 1). These results suggested an enhanced ROS scavenging capability of *Osaca9*, which helps to prevent the excessive accumulation of ROS from causing damage to plant cells while improving disease resistance. Previous studies have also reported some of the ACAs’ roles in ROS production and scavenging [35,40]. *OsACA6* was reported to efficiently modulate ROS metabolism and proline biosynthesis in response to the abiotic stresses of salt, drought, cold, and heat in tobacco [40,41]. Thus, we speculated that OsACA9 might also have a role in stress response, which must be investigated through further studies. However, the excessive accumulation of ROS has also proved harmful to organisms as it triggers protein oxidation and enzyme inactivation and leads to reduced photosynthesis efficiency and impaired chloroplast development [45,46]. 

In addition, *Osaca9* mutants displayed an early-leaf-senescence phenotype, especially during the filling stage (Figure 3D and Figure 4A–C). We speculated that the early-leaf-senescence phenotype was partly affected by ROS accumulation, as an expense of its enhanced resistance. Further, changes in endogenous phytohormone levels influence signaling networks that function in senescence processes. Plant hormones such as ethylene, jasmonic acid (JA), salicylic acid (SA), abscisic acid (ABA), and brassinosteroid (BR) promote leaf senescence and are extensively involved in responses to various abiotic and biotic stresses [72]. According to the transcription analysis, the upregulation of *EIN3* (Figure 6C), a key transcription factor in the ethylene response pathway which upregulates the transcription level of *ORE1* by inhibiting the transcription of miR164 [73], and the downregulation of *BAK1* (Figure 6C), an important factor in BRI1-mediated BR signaling, may have contributed to the early senescence of *Osaca9*. Further, master TFs-mediated transcriptional regulation plays a crucial role in the regulation of leaf senescence [74]. Members of WRKY TFs and the basic helix-loop-helix (bHLH) family TFs, which coordinate with endogenous hormones to finely regulate the leaf senescence process, were upregulated in *Osaca9* (Table 1, Appendix A). Moreover, RKs and MAPKs, regulating leaf senescence by affecting the phosphorylation status of target proteins [75], were also detected in DEGs (Figure 6B,C, Table 1). These results finely explain the early-leaf-senescence phenotype of *Osaca9*.

In addition, the increased resistance might also cause dwarfism, through shortened leaves and panicles [76]. According to the statistical analysis of agronomic traits of the *Osaca9* mutant, unsurprisingly, the length and width of the flag leaves of knockout lines decreased significantly. The length of the main panicle became shorter, leading to a decreased grain number (Figure 5A,B,D,E,G,H). However, there was no significant difference in seed setting rate and 1000-grain weight between *Osaca9* and wild-type (Figure 5J,N), suggesting that *Osaca9* did not affect grain filling and seed production. Interestingly, the reduced yield per plant of *Osaca9* was partly compensated by increased available tiller number (Figure 5I), suggesting that *OsACA9* might have an additional role in regulating the rhizomes of rice.

In plants, Ca^2+^ is an important nutrient and cellular secondary signaling molecule that is essential for responses to biotic and abiotic stimuli as well as growth and development [11,77]. Owing to its cytotoxicity, cytosolic Ca^2+^ levels must be maintained at low (10^−8^ to 10^−7^ M) levels in living cells [78]. Ca^2+^ homeostasis is maintained by an array of Ca^2+^ transport elements during unfavorable situations including Ca^2+^-ATPases, cyclic nucleotide-gated channels (CNGCs), and glutamate receptor homologs (GLRs), etc. [79]. ACAs are considered one major type of Ca^2+^-ATPases that mediate active Ca^2+^ transport out of the cytosol. In *OsACA9* knockout mutants, the efflux of Ca^2+^ might be hindered, resulting in the continuous accumulation of free cytosolic Ca^2+^. The cytosol Ca^2+^ burst has been proven to promote the phosphorylation of RBOH and trigger ROS production [80], as well as the upstream activation of many responses, including the activation of MAPK kinases, as well as the induction of pathogenesis-related gene expression [81], leading to enhanced resistance in *Osaca9* mutants. It can be implied that OsACA9 might balance plant growth and development with immunity response in such model (Figure 6D). In addition, a deeper understanding of the growth–defense trade-offs mediated by OsACA9 will be a powerful tool to reveal plant–bacterial interaction and can be useful for optimizing crop breeding [79]. We are currently conducting a detailed study to determine how mutations in *OsACA9* activate pivotal immune pathways, especially the MAPK signaling pathway, and further explore the molecular mechanisms of how Ca^2+^-ATPase orchestrates external stimulus signaling through the Ca^2+^ signature.

## 4. Materials and Methods

### 4.1. Plant Materials

A *Japonica* rice cultivar named ZH11 was used for knockout of *OsACA9* gene by CRISPR/Cas9. The different knockout lines of *OsACA9* gene, *aca9#1* and *aca9#2*, and the wild-type control ZH11 were planted in the field of an experimental farm in summer in South China. For the plot yield test, 200 individuals were planted in an 8 m^2^ (2 × 4) plot with a planting density of 20 cm × 20 cm. The agronomic traits of rice plants were measured on individual plants with at least 10 replicates at the mature stage.

### 4.2. Phylogenetic Analysis of OsACA Gene Family

The *OsACA* gene family has been reported in rice by Singh et al. (2014). For phylogenetic analysis, protein sequences of ACA family members of rice (*O. sativa*), Arabidopsis (*A. thaliana*) [37], tomato (*S. lycopersicum*) [17], and maize (*Z. mays*) [20] were used, and a phylogenetic tree was constructed by the neighbor joining method with 1000 bootstrap replicates by MEGA7.0 [82]. *OsACA* genes’ structure and clutter were performed using TBtools [83].

### 4.3. Construction of OsACA9 Knockout Rice Plants by CRISPR/Cas9

The coding sequence (CDS) of the *OsACA9* gene (*Os02g0176700*) was analyzed by the program CRISPR Design: http://crispr.mit.edu (accessed on 28 June 2022). Two sgRNAs, sgRNA-1 (5′-GCTTCATATGAAATTTGCGG-3′) and sgRNA-2 (5′-ATACAGTGACGCTGAAAGGG-3′), were selected for *OsACA9* gene editing and constructed into a CRISPR/Cas9 vector using the BGK03 kit (Biogle, Changzhou, China) according to the manufacturer’s instructions. The vector was transformed into the wild-type rice cultivar ZH11 by the Agrobacterium-mediated method [84]. The genomic DNA from the leaves of transgenic plants was extracted using the CTAB method and used for PCR amplification of *OsACA9* gene editing target sites. The primer pairs used for PCR were sgRNA-1 F: 5′-CATGCAGGGGAAGCGTTT-3′ and sgRNA-1 R: 5′-CCTTTTGCCAATCAACCA-3′, and sgRNA-2 F: 5′-GCTCCAGTGAACATAAGATA-3′ and sgRNA-2 R: 5′-CTCGAAAGTATGTACTCCAGAT-3′.

### 4.4. Assay of Chlorophyll Content and Photosynthesis

At the filling stage, 0.1 g of fresh leaves of rice plants were sampled, and the chlorophyll pigment was extracted with equal volumes of 80% acetone. All samples were kept in a dark place at room temperature for 24 h with frequent shaking until all leaves turned white. The absorbance of the supernatant was measured by the spectrophotometer NanoDrop2000C (Thermo Fisher Scientific, Wilmington, DE, USA) at 665 nm, 649 nm, and 470 nm. Each sample was tested with at least three biological repeats. The concentrations of chlorophyll a, chlorophyll b and carotenoids were calculated according to Lichtenthaler’s method [85]. For the assay of photosynthesis, the net photosynthetic rate of the flag leaves of rice plants was measured using the LI-6400 Portable Photosynthesis System (LI-COR, Lincoln, NE, USA) following the manufacturer’s instructions.
Chlorophyll a concentration = 13.95*A*_665_ − 6.88*A*_649_
Chlorophyll b concentration = 24.96*A*_649_ − 7.32*A*_665_
Carotenoid concentration = (1000*A*_470_ − 2.05Chla − 114Chlb)/245

### 4.5. Disease Resistance Identification of Rice Plants

Six *Xoo* strains (PXO86, PXO71, POX99A, Zhe173, C2 and C5) and two *Xoc* strains (BLS256 and RS105) were used for disease resistance identification. Strains were cultured on a potato dextrose agar (PDA) medium that contained 20 g sucrose, 5 g peptone, 0.5 g Ca(NO_3_)_2_, 0.43 g Na_2_HPO_4_, and 0.05 g FeSO_4_ per liter and grew at 28 °C for 3 days. The bacterial colony was suspended in sterile distilled water at an optical density of OD_600_ = 1.0 and immediately used for inoculation. At the tillering stage, the leaf tip clipping method [86] was used for inoculation of *Xoo*, and the non-needle syringe injection method [87] was used for inoculation of *Xoc*. Lesions on the inoculated leaves were measured for the evaluation of resistance 2 weeks after inoculation.

### 4.6. DAB Staining and H_2_O_2_ Assay

Leaves of the rice plant were stained by DAB according to Thordal-Christensen’s method [88]. In brief, leaves were placed in 1 mg/mL DAB–HCI and 10 mM ascorbic acid, pH 3.8, incubated in the growth chamber for 8 h, and then cleared in boiling ethanol (96%) for 10 min. H_2_O_2_ is visualized as a reddish brown coloration. Peroxide (H_2_O_2_) Content Assay kits (Solarbio, Beijing, China) were used to measure endogenous H_2_O_2_ content. Each measurement was performed in at least three biological replicates.

### 4.7. RNA-Seq and Data Analysis

At the tillering stage, leaves of rice plants were sampled and sent to the Beijing Genomics Institute (Shenzhen, China) for the construction of the cDNA library and next-generation sequencing (NGS). Each sample had three biological replicates which were sequenced by the BGISEQ-500 platform. Differentially expressed genes (DEGs; log_2_ ratio ≥1 or ≤−1) were detected with a significance *p* < 0.05 between samples. Gene functional enrichments were analyzed using GO: http://www.geneontology.org/ (accessed on 11 August 2023) and KEGG: http://www.genome.jp/kegg/ (accessed on 11 August 2023). 

### 4.8. RNA Extraction and RT-qPCR

Total RNA of rice was extracted using RNeasy Plant Mini Kit (QIAGEN, Valencia, CA, USA) and RNase-Free DNase Set (QIAGEN, Valencia, CA, USA) following the manufacturer’s instruction. Synthesis of first-strand cDNAs from RNA was carried out by the M-MLV Reverse Transcriptase (Promega, Beijing, China) according to the manufacturer’s instructions. qPCR was conducted using TB Green™ Premix Ex *Taq*™ (TaKaRa, Beijing, China) with a final volume of 10 μL per reaction. Each reaction mixture consisted of 5.0 μL TB Green Premix Ex Taq (TaKaRa, Beijing, China), 1.0 μL cDNA template, 0.5 μL of each primer, and 3.3 μL RNase-Free double distilled water. The qPCR was performed on the Step OnePlus™ Real-Time PCR System (Applied Biosystems, Foster City, CA, USA) using the following program: 95 °C for 5 min, followed by 40 cycles at 95 °C for 5 s and 60 °C for 30 s. Relative expression level of the interest gene was calculated by the 2^−ΔΔCT^ method using the housekeeping genes *OsEF1* and *OsActin* as the standardization controls [89]. Each measurement was performed in at least three biological replicates. The primers are listed in Table 2.

## Figures and Tables

**Figure 1 ijms-25-01874-f001:**
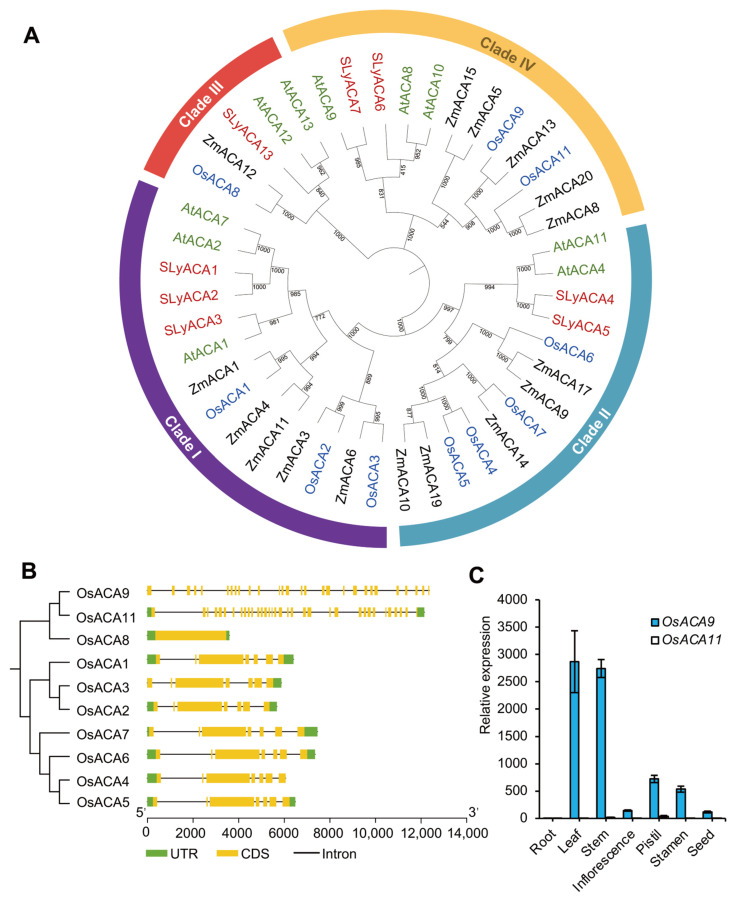
Phylogenetic tree of the OsACA family members and their expression in rice tissues. (**A**) Phylogenetic relationship between ACA proteins from *O. sativa* (blue), *A. thaliana* (green), *S. lycopersicum* (red), and *Z. mays* (black). (**B**) Cluster and gene structure of *OsACAs*. The neighbor-joining tree was constructed using 1000 bootstrap replicates. (**C**) Expression pattern of *OsACA9* and *OsACA11* in rice different tissues by RT-qPCR.

**Figure 2 ijms-25-01874-f002:**
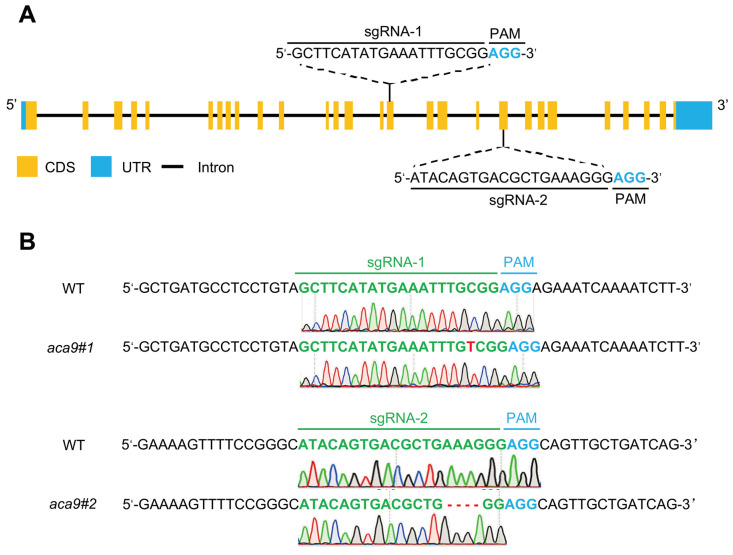
The sgRNAs and gene editing target sites of *OsACA9* by CRISPR-Cas9 in rice. (**A**) The sequences and target sites of the two different sgRNAs, sgRNA-1 and sgRNA-2, in the *OsACA9* gene. The sgRNA sequences of the target are shown in black and the sequences of PAM (protospacer-adjacent motif) are shown in blue. (**B**) The sequencing results of the two sgRNA targets in the *OsACA9* gene from different mutation lines.

**Figure 3 ijms-25-01874-f003:**
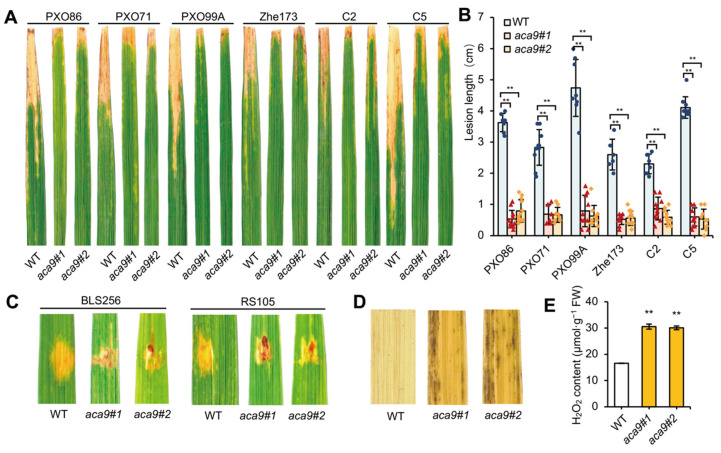
Resistance evaluation of *Osaca9* to *Xanthomonas oryzae* pv. *oryzae* (*Xoo*) and *Xanthomonas oryzae* pv. *oryzicola* (*Xoc*) strains and ROS detection in the *Osaca9* mutants. (**A**–**C**) Defense reactions of wild-type (ZH11) and the *Osaca9* mutants to *Xoo* strains (PXO86, PXO71, POX99A, Zhe173, C2 and C5) (**A,B**) and *Xoc* strains (BLS256 and RS105) (**C**). (**D**) DAB staining of leaves of wild-type and the *Osaca9* mutants. (**E**) Statistical analysis of H_2_O_2_ content. WT refers to the wild-type control, and *aca9#1* and *aca9#2* refer to two distinct knockout lines of *OsACA9*. ** indicates the significant difference at *p* < 0.01 by *t*-test.

**Figure 4 ijms-25-01874-f004:**
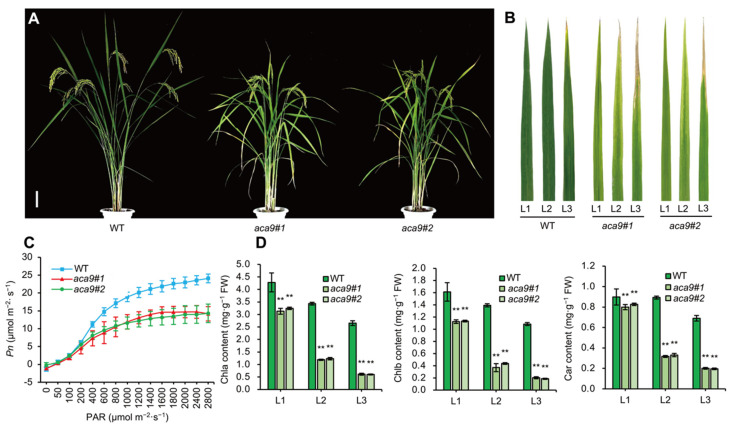
Early-leaf-senescence phenotype of the *Osaca9* mutants. (**A**) Plant phenotype at filling stage (bar = 10 cm). (**B**) The flag leaf (L1), second top leaf (L2), and third top leaf (L3) at filling stage. (**C**) Light-response curve of net photosynthesis rate (*Pn*) of the flag leaves at filling stage. (**D**) Chlorophyll a (Chla), Chlorophyll b (Chlb), and Carotenoid (Car) content of the leaves shown in (**B**). WT refers to the wild-type control, and *aca9#1* and *aca9#2* are two distinct knockout lines of the *OsACA9* gene. ** indicates the significant difference at *p* < 0.01 by *t*-test.

**Figure 5 ijms-25-01874-f005:**
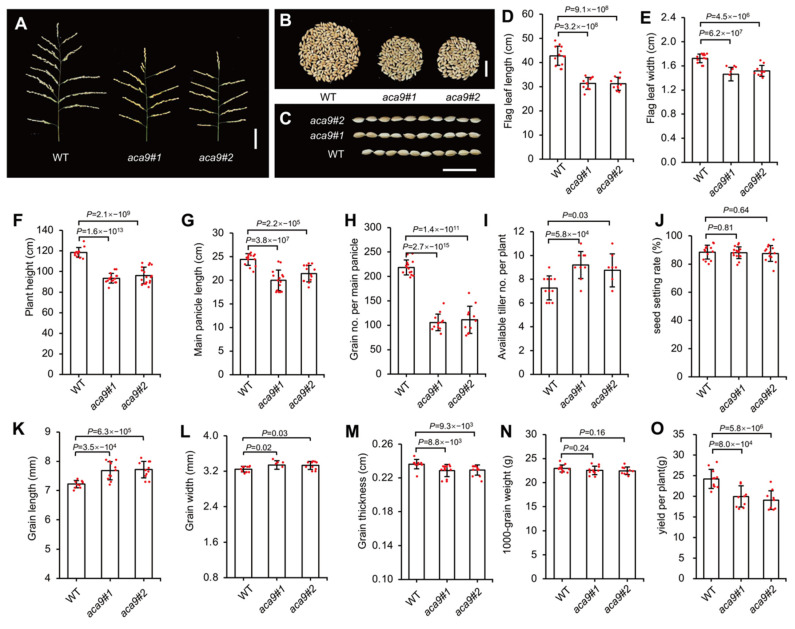
Agronomic characters of the *Osaca9* mutants. (**A**) Pictures of main panicles (bar = 5 cm). (**B**) All grains per panicle (bar = 2 cm). (**C**) Length of ten grains (bar = 2 cm). (**D**) Flag leaf length. (**E**) Flag leaf width. (**F**) Plant height. (**G**) Main panicle length. (**H**) Grain number per main panicle. (**I**) Effective tiller number per plant. (**J**) Seed setting ratio. (**K**) Grain length. (**L**) Grain width. (**M**) Grain thickness. (**N**) 1000-grain weight. (**O**) Yield per plant. The number on the histogram indicates the *p* value of the significance between *aca9#1* or *aca9#2* and the wild-type control (WT) by *t*-test.

**Figure 6 ijms-25-01874-f006:**
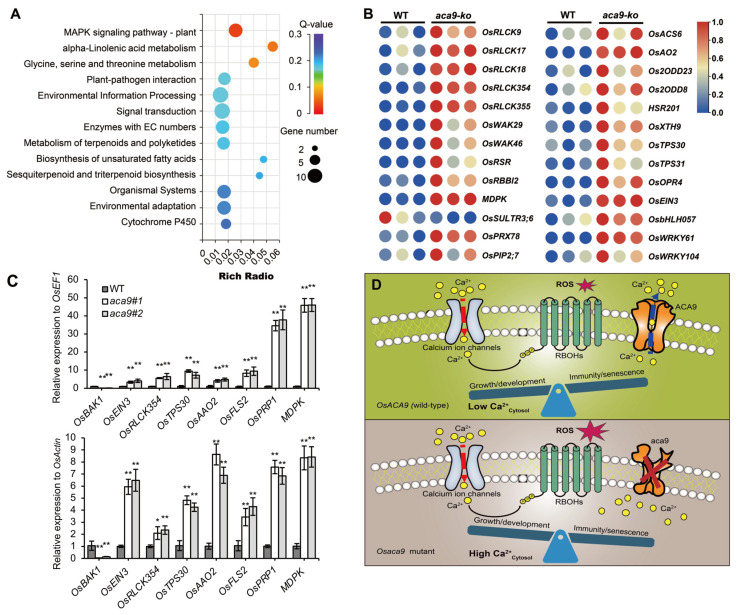
Transcriptome analysis of the *Osaca9* mutants and the schematic diagram of the predicted mechanism mediated by *OsACA9*. (**A**) KEGG pathway enrichment analysis of the differentially expressed genes (DEGs) between the *Osaca9* mutants and the wild-type (WT) control. (**B**) Heatmap of the relative expression levels of representative DEGs involved in plant immunity. (**C**) Expression analysis of disease resistance-associated genes between WT and the *Osaca9* mutation lines, using *OsEF1* and *OsActin* as the standardization controls, respectively. (**D**) The schematic diagram of the predicted disease resistance mechanism mediated by *OsACA9*. *Aca9#1* and *aca9#2* are two distinct knockout lines of the *OsACA9* gene. * and ** indicate the significance at *p* < 0.05 and *p* < 0.01 by *t*-test, respectively.

**Table 1 ijms-25-01874-t001:** Differentially expressed genes (DEGs) related to disease resistance between the *Osaca9* mutants and the wild-type (WT) control.

Gene ID	Gene Name	Definition	Fold (log2)	Function	Reference
*Os01g0113500*	*OsRLCK9*	receptor-like kinase	2.642		
*Os01g0116900*	*OsRLCK17*	LRK-type protein; protein kinase domain containing protein	3.050	Resistance to *Xanthomonas oryzae*	[49]
*Os01g0116800*	*OsRLCK18*	TAK14	4.914	Resistance to abiotic stress	[49]
*Os11g0695000*	*OsRLCK354*	leucine-rich repeat receptor protein kinase EXS precursor	6.877		
*Os11g0691800*	*OsRLCK355*	receptor-like protein kinase precursor	4.635		
*Os04g0127500*	*OsWAK29*	OsWAK receptor-like protein kinase	6.259		
*Os04g0370100*	*OsWAK46*	OsWAK receptor-like protein kinase	7.100		
*Os02g0618200*	*OsRSR1*	two-component response regulator	3.034	Regulate tiller growth and panicle development	[50]
*Os11g0229500*	*OsRSR*	disease resistance protein	9.878	Resistance to SHB	[51]
*Os01g0124000*	*OsRBBI2*	BBTI2-Bowman-Birk type bran trypsin inhibitor precursor	6.995		
*Os09g0355400*	*MDPK*	malectin domain protein kinase	9.393	Resistance to SHB	[52]
*Os01g0719300*	*OsSULTR3;6*	secondary active sulfate transmembrane transporter activity	−4.383	Resistance to bacterial leaf streak	[53]
*Os06g0306300*	*OsPRX78*	peroxidase precursor	3.878		
*Os09g0541000*	*OsPIP2;7*	aquaporin, plasma membrane intrinsic protein	4.199	Tolerance to boron toxicity	[54]
*Os01g0901600*	*OsACS6*	AMP-binding domain containing protein	2.457	Phenylpropanoid biosynthesis	
*Os04g0182200*	*Os2ODD23*	1-aminocyclopropane-1-carboxylate oxidase homolog 2	2.366	Benzoxazinoid biosynthesis	[55]
*Os03g0289800*	*Os2ODD8*	leucoanthocyanidin dioxygenase	2.387	Response to low-temperature stress and salt stress	[56]
*Os04g0604300*	*OsXTH9*	glycosyl hydrolases family 16	3.210	Cell wall modification	[57]
*Os08g0167800*	*TPS30;46*	terpene synthase	4.295	Key gene for biosynthesis of limonene, methyl salicylate	[58]
*Os08g0168000*	*OsTPS31*	terpene synthase activity	4.549	Resistance to insect	[59]
*Os07g0543000*	*OsbHLH057*	bHLH transcription factor	1.557	Stress tolerance and yield	[60]
*Os06g0215900*	*OsOPR4*	12-oxophytodienoate reductase	8.469	Plant hormone response related; Jasmonic Acid biosynthesis related	[61]
*Os03g0324200*	*OsEIN3*	ethylene signaling regulation factors	6.398	Involved in environmental humidity regulation of rice blast resistance in rice	[62]
*Os11g0117400*	*OsWRKY104*	WRKY transcription factor	2.929	Response to biotic and abiotic stresses	[63]
*Os11g0685700*	*OsWRKY61*	DNA-binding transcription factor activity	9.098		
*Os12g0458100*	*HSR201*	transferase family protein, putative, expressed	5.811	Hypersensitivity-related; required for pathogen signal-induced salicylic acid synthesis	[64]
*Os11g051450*	*OsBAK1*	brassinosteroid insensitive 1-associated receptor kinase 1 precursor	−6.408	plant growth and development; PAMP triggered immune response	[65]
*Os10g0138100*	*OsAO2, AAO2*	aldehyde oxidase	12.345	Regulates plant growth, grain yield, and drought tolerance	[66]
*Os04g0618700*	*OsFLS2*	leucine-rich repeat receptor protein kinase EXS precursor; flg22 Receptor	2.101	PAMP triggered immune response	[67]

**Table 2 ijms-25-01874-t002:** Primers used for qPCR.

Gene	Forward Primer (5′–3′)	Reverse Primer (5′–3′)
*OsActin*	GTTACTCATTCACCACAACGGC	CCTTTCAGGAGGGGCGACC
*OsEF1*	AAGAGGAAGTCAGCGGCTAAG	CAGAATGGGCAGGAAAATACA
*OsACA9*	CGGGACGACGACGATGGC	CGCGCTTGGCGGGGATGT
*OsACA11*	TGGCGGCAAGCAGCTCTAGT	CGGAATGCAGCCCTGACAAC
*OsBAK1*	ATCAACTGGGTGGAGAGTGA	ATCTCCCAAGGTGTGTGGTA
*OsEIN3*	CCTCAAGAAGGCCTGGAAGG	CTTGGCGGTCATCTTGTCCT
*Os* *RLCK354*	TCCGTCCTTAACCTCACCAT	GGGATTTGTCCAGAGAGCTG
*OsTPS30*	TACCGCTGGCTATACTCACA	CTCATGGCTTCTAGGCTTGG
*OsAO2*	TGCTGGGAGAAAGCCGAAAT	TGGCACTCTGCACTTGACTT
*OsFLS2*	AGAGATTGTCGCTCCATGCC	AGAGCGAGTTGTTCTGGACG
*OsRSR1*	AACACACTCCCAATCCGCTT	AGGGTGGTTTTCCCCAATCC
*MDPK*	GGACGCAACACCTGGATACT	CGTGTATGGAGTGGTGCCAT

## Data Availability

The raw data of transcriptomic sequencing can be downloaded in the NCBI Sequence Read Archive under accession number PRJNA1056136.

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
