# Peer review of "OsACA9, an Autoinhibited Ca2+-ATPase, Synergically Regulates Disease Resistance and Leaf Senescence in Rice"

_ijms, 2024, doi:10.3390/ijms25031874_

Round 1

Reviewer 1 Report

Comments and Suggestions for Authors

The article entitled, “OSACA9, an Autoinhibited CA2+-ATPase Synergically Regulates Disease Resistance and Leaf Senescence in Rice” submitted by Dr. Xinyu Wang, can be accepted after major revision. It needed to be improved in areas like grammatical and scientific writing. Introduction needs to be updated with the recent reports on calcium signaling in the para one. Author may see PMID: 37875477 and 36611920. About CA2+-ATPase can be further improved in the para 2 of introduction as described in the Elsevier book Calcium transport elements in plants. In the para 3, authors have mentioned CA2+-ATPase of several plant species but missed the bread wheat PMID: 29792165, and other monocot, which are more important than the dicot species. What is the meaning of 2.1. Evolution analysis suggesting OsACA9 may biologically function in rice? It needs to be clarified. Authors should provide sequencing info for more lines. Some of the data of transcriptome sequencing should be validated by qRT PCR. Discussion should be focused on the major finding, avoid any repetitions from the result section. Method sections should be elaborated. Some sections have just cited a ref., atleast basic details should be elaborated. Conclussion may contain the future perspective broadly. Suggested minor revision- 1. There are many grammatical and spelling errors throughout the manuscript which need to be corrected after thorough reading. 2. A few lines mentioned below are examples which should be also checked and corrected after thorough revision- Line no. 2: In the title, “OSACA9” which should be “OsACA9” as used in the whole manuscript. Line no. 138: Full stop after completion of sentence. Line no. 141: Shew, not commonly used in scientific writing. Line no. 198-199: Formatting error. 3. Also, 1- In Figure 6 (C), the spelling of relative is written relative. 2- In Figure 6 (D), the spelling of wild type is wide type.

Comments on the Quality of English Language

Corrections required

Author Response

Response to the comments of Reviewer#1:

Comment 1: Introduction needs to be updated with the recent reports on calcium signaling in the para one. Author may see PMID: 37875477 and 36611920. About Ca2+-ATPase can be further improved in the para 2 of introduction as described in the Elsevier book Calcium transport elements in plants. In the para 3, authors have mentioned Ca2+-ATPase of several plant species but missed the bread wheat PMID: 29792165, and other monocot, which are more important than the dicot species.

Reply: Thanks for your constructive suggestion, both the introduction and discussion have been rewritten and all mentioned references have been added in our revised manuscript. Just as you mentioned, monocot plants are important food crops, and to reveal the function of Ca2+-ATPase in monocot is more meaningful, however, up to now, studies on the Ca2+-ATPase of monocot were mainly focus on genetic analysis, so here we only added related functional advances in this part.

Comment 2: What is the meaning of 2.1. Evolution analysis suggesting OsACA9 may biologically function in rice? It needs to be clarified.

Reply: In 2.1, we are trying to explore the classification and evolution of the ACA family in rice, and explain why we chose OsACA9 for functional research. According to the function of the homologue of OsACA9 in Arabidopsis we speculated that OsACA9 might play an potential role in defense response, as well as the growth and development in rice. To make it clearer, we have changed the subtitle into ‘2.1 Evolution analysis and expression profile suggest the potential role of OsACA9 in various biological processes’ and described the conclusion at the end of 2.1.

Comment 3: Authors should provide sequencing info for more lines.

Reply: When creating gene editing materials, we designed two different targets on the CDS of OsACA9 gene, and for each target, the phenotype of all mutation lines were absolutely identical , so we only selected one individual T0 mutation line from each target for subsequent research, including identification of both disease resistance and phenotype.

Comment 4: Some of the data of transcriptome sequencing should be validated by qRT PCR.

Reply: According to your suggestion, we carried out qRT-PCR of several different expression genes whose function have been reported and proved to play a vital role in plant immunity (Figure 6C).

Comment 5: Discussion should be focused on the major finding, avoid any repetitions from the result section.

Reply: Thanks for your advice, according to your suggestion, we have rewritten the relevant content and highlighted in the discussion part.

Comment 6: Method sections should be elaborated. Some sections have just cited a ref., at least basic details should be elaborated.

Reply: According to your suggestion, more details have been added in the method part from 4.3 to 4.7.

Comment 7: Conclusion may contain the future perspective broadly.

Reply: Thanks for your suggestion, we have added future perspectives related to reveal the function of both OsACA9 and other autoinhibited type Ca2+-ATPase at the end of discussion.

Suggested minor revision

There are many grammatical and spelling errors throughout the manuscript which need to be corrected after thorough reading. A few lines mentioned below are examples which should be also checked and corrected after thorough revision- Line no. 2: In the title, “OSACA9” which should be “OsACA9” as used in the whole manuscript. Line no. 138: Full stop after completion of sentence. Line no. 141: Shew, not commonly used in scientific writing. Line no. 198-199: Formatting error. [thanks for your careless check, the error has been rectified] 3. Also, 1- In Figure 6 (C), the spelling of relative is written relative. 2- In Figure 6 (D), the spelling of wild type is wide type. [thanks for your careless check, the word has been rectified]

Reply: Thanks for your careful check, all the words mentioned has been rectified and we have checked the whole manuscript for grammatical and spelling errors after thorough reading.

Reviewer 2 Report

Comments and Suggestions for Authors

I commend your meticulous investigation into the role of the Autoinhibited Ca2+-ATPase OsACA9 in rice. Your systematic analysis of the OsACA family and the subsequent CRISPR/Cas9 knockout of OsACA9 provide valuable insights into its multifaceted functions in cellular Ca2+ homeostasis, growth, stress tolerance, and immune response in plants. The observed increase in reactive oxygen species (ROS) accumulation in the OsACA9 knockout lines, coupled with enhanced disease resistance to bacterial blight and leaf streak, underscores the pivotal role of OsACA9 in mediating plant defense mechanisms. The identification of altered agronomy traits and the early leaf senescence phenotype further contributes to our understanding of OsACA9's broader impact on plant physiology. Your comprehensive transcriptome analysis sheds light on the molecular pathways involved, emphasizing OsACA9's influence on basal immune pathways and antibacterial metabolite synthesis. This research significantly advances our knowledge of the Ca2+-ATPase OsACA9 and its intricate involvement in orchestrating disease resistance and leaf senescence in rice.

Author Response

Response to the comments of Reviewer#2:

Comment: I commend your meticulous investigation into the role of the Autoinhibited Ca2+-ATPase OsACA9 in rice. Your systematic analysis of the OsACA family and the subsequent CRISPR/Cas9 knockout of OsACA9 provide valuable insights into its multifaceted functions in cellular Ca2+ homeostasis, growth, stress tolerance, and immune response in plants. The observed increase in reactive oxygen species (ROS) accumulation in the OsACA9 knockout lines, coupled with enhanced disease resistance to bacterial blight and leaf streak, underscores the pivotal role of OsACA9 in mediating plant defense mechanisms. The identification of altered agronomy traits and the early leaf senescence phenotype further contributes to our understanding of OsACA9's broader impact on plant physiology. Your comprehensive transcriptome analysis sheds light on the molecular pathways involved, emphasizing OsACA9's influence on basal immune pathways and antibacterial metabolite synthesis. This research significantly advances our knowledge of the Ca2+-ATPase OsACA9 and its intricate involvement in orchestrating disease resistance and leaf senescence in rice.

Reply: Thank you for your careful review and approval for our work, we will conduct in-depth research on the functional mechanism of OsACA9 in the future, in order to provide reference for the specific mechanism of this type of Ca2+-ATPase in plant immunity and development, and its signaling pathways for maintaining calcium ion balance.

Reviewer 3 Report

Comments and Suggestions for Authors

ijms-2830858

Overview:

This manuscript describes the biological functions of OsACA9 Ca2+-ATPase in rice. The experimental design is enough to draw the conclusion, but some descriptions are insufficient.

The current standard RT-qPCR protocol requires two housekeeping genes and three technical replicates for each biological replicate.

Discussion on senescence is insufficient. What are the expression levels of senescence-related genes?

Some important information is missing. Which tissues at which developmental stages were subjected to transcriptome analysis? Does the three rows of colored circles in Fig. 6B indicate the three replicates?

Please correct typographical and careless mistakes in the manuscript.

Individual comments:

L15: cluttered à clustered

Figure 1A: The border between Clade I and II is incorrect.

L141, L193: shew à showed?

L190: receptively à respectively?

L199: Something is missing.

L258, L260: (Figure 5) à (Figure 6)

L275: [32, 37, 50] incorrect insertion

Author Response

Response to the comments of Reviewer#3:

Comment 1: The current standard RT-qPCR protocol requires two housekeeping genes and three technical replicates for each biological replicate.

Reply: Thanks for your constructive suggestion. We have added the housekeeping gene of OsEF1 as another standardization control, and the result were basically consistent with the expression trend when utilization OsActin as the standardization control (Figure 6C).

Comment 2: Discussion on senescence is insufficient. What are the expression levels of senescence-related genes?

Reply: Thanks for your reminder. We have elaborated the part related to senescence in discussion. In our RNA-Seq, different expression genes related to senescence were not enriched between WT and the aca9 mutation lines. For this result, we speculate that the senescence phenotype of aca9 mutant may be indirectly caused by the excessive accumulation of calcium ions in cells, rather than directly regulating the expression of aging process.

Comment 3: Some important information is missing. Which tissues at which developmental stages were subjected to transcriptome analysis?

Reply: The sample of RNA-Seq is taken at tillering stage, and we have added the stage detail in 4.7. The raw data and detail information of transcriptomic sequencing can be downloaded in the NCBI Sequence Read Archive under accession number PRJNA1056136.

Comment 4: Do the three rows of colored circles in Fig. 6B indicate the three replicates?

Reply: Yes, each circle indicates the normalized relative expression level of a biological replicate.

Comment 5: Please correct typographical and careless mistakes in the manuscript.

Reply: Thanks for your careful check, all the authors have rechecked the whole manuscript for grammatical and spelling errors after thorough reading, and made corresponding revision in the revised manuscript with highlight.

Comment 6: L15: cluttered à clustered. Figure 1A: The border between Clade I and II is incorrect. L141, L193: shew à showed? L190: receptively à respectively? L199: Something is missing. L258, L260: (Figure 5) à (Figure 6) L275: [32, 37, 50] incorrect insertion.

Reply: Thanks for your careful check, the above mistakes mentioned have been rectified in the revised manuscript and highlighted in red.

Round 2

Reviewer 1 Report

Comments and Suggestions for Authors

The Ms has been significantly improved. Moreover, the concluding para may be further improved. Authors should see the latest book "Calcium transport elements in plants" published by Elsevier to modify. Further, its last chapter has discussed numerous applications, that may also be discussed.

Comments on the Quality of English Language

 Minor editing of English language required

Author Response

Response to the comments of Reviewer#1:

Comment 1: The Ms has been significantly improved. Moreover, the concluding para may be further improved. Authors should see the latest book "Calcium transport elements in plants" published by Elsevier to modify. Further, its last chapter has discussed numerous applications, that may also be discussed.

Reply: Thank you for your suggestions and recommendations for excellent books on calcium transport, which greatly enriches our understanding of the relevant field. According to your advice, the concluding para has been further elaborated, we addded more detailed information on Ca2+ transport in plant cells and made a prospect of the role of OsACA9 gene. Meanwhile, we also added more information relative to Ca2+, ROS and immunity in the introduction part.

Comment 2: Minor editing of English language required.

Reply: Thanks for your reminder. All of our authors have conducted a comprehensive check and correction of grammar and writing errors throughout the manuscript, all of those modification were highlighted in the revised manuscript with red font.

Reviewer 3 Report

Comments and Suggestions for Authors

Your manuscript has been revised satisfactorily.

Author Response

Response to the comments of Reviewer#3:

Comment 1: Your manuscript has been revised satisfactorily.

Reply: Thank you again for your careful review and approval, which are very valuable and helpful for improving the manuscript and our work.